# Bimetallic Nanoparticles as a Model System for an Industrial NiMo Catalyst

**DOI:** 10.3390/ma12223727

**Published:** 2019-11-12

**Authors:** Sara Blomberg, Niclas Johansson, Esko Kokkonen, Jenny Rissler, Linnéa Kollberg, Calle Preger, Sara M Franzén, Maria E Messing, Christian Hulteberg

**Affiliations:** 1Department of Chemical Engineering, Lund University, 221 00 Lund, Sweden; 2Advanced Light Source, Lawrence Berkeley National Laboratory, Berkeley, CA 94720-8196, USA; 3MAX IV Laboratory, Lund University, 221 00 Lund, Sweden; 4Bioscience and Materials, RISE Research Institute of Technology, 223 70 Lund, Sweden; 5Design Sciences, Lund University, 221 00 Lund, Sweden; 6SunCarbon, 223 62 Lund, Sweden; 7NanoLund, Division of Solid State Physics, Lund University, 221 00 Lund, Sweden

**Keywords:** NiMo catalyst, alloyed nanoparticles, model catalyst, industrial catalysts, lignin

## Abstract

An in-depth understanding of the reaction mechanism is required for the further development of Mo-based catalysts for biobased feedstocks. However, fundamental studies of industrial catalysts are challenging, and simplified systems are often used without direct comparison to their industrial counterparts. Here, we report on size-selected bimetallic NiMo nanoparticles as a candidate for a model catalyst that is directly compared to the industrial system to evaluate their industrial relevance. Both the nanoparticles and industrial supported NiMo catalysts were characterized using surface- and bulk-sensitive techniques. We found that the active Ni and Mo metals in the industrial catalyst are well dispersed and well mixed on the support, and that the interaction between Ni and Mo promotes the reduction of the Mo oxide. We successfully produced 25 nm NiMo alloyed nanoparticles with a narrow size distribution. Characterization of the nanoparticles showed that they have a metallic core with a native oxide shell with a high potential for use as a model system for fundamental studies of hydrotreating catalysts for biobased feedstocks.

## 1. Introduction

The depletion of fossil fuels and the effects of global warming underline the importance of finding alternative fuels in the near future. Renewable feedstocks have been the subject of considerable attention in recent decades, and a large number of potential renewable resources, such as lignin, have been studied. Lignin is the second most abundant natural polymer in nature and is therefore believed to have considerable potential as a value-added feedstock [1,2]. Catalytic hydrotreating processes using traditional Mo-based catalysts promoted by Ni or Co have been found to be promising in upgrading lignin and other biomass-based feedstocks into high-quality fuel [3,4,5,6]. Mo-based catalysts have played an essential role in the hydrotreating reaction in the petrochemical industry for several decades and have, therefore, been studied in in-depth [7], however, we still do not have a complete understanding of the active site of the bimetallic catalyst. It has been suggested that Ni or Co promote the activity of Mo catalysts by affecting the reactivity of the Mo atoms, but fundamental studies on an atomic level are required to obtain detailed information on the impact of promotors in Mo catalysts. Industrially produced catalysts are complex, with a low loading of the active Mo and Ni metals and preferably, uniformly dispersed over a porous oxide support, resulting in a high surface-to-volume ratio of the active metals. The uniform dispersion of the metals makes characterization on the atomic level challenging, and simplified model systems are often used in detailed studies of the interaction between the reactants and the active surface of the catalyst [8,9]. Studies of simplified model systems using single crystals have shown that the active site is associated with the edges of the sulfidic phase of Mo forming 2-dimensional S–Mo–S layer structures in which the basal planes are inactive [10]. Even though the active phase is the sulfide, the industrial catalysts are prepared in the oxide form and it has been shown that the structure of the oxide has a significant impact on the formation of the sulfide. A detailed understanding of the oxide precursor is therefore of importance for the final catalyst performance [11]. Previous studies of bimetallic catalysts have shown significantly increased complexity of the atomic structure when Ni or Co is added to Mo catalysts as a promotor, but the results of these studies also suggest that the promotors are associated with the edges of the S–Mo–S structures [12,13]. The above-mentioned studies highlight the importance of edges and corners in the activity of Mo-based catalysts, indicating that nanoparticles would be suitable as simplified model systems, as they have a high density of edges and corners relative to their surface area [14]. 

To produce relevant model systems, the properties of the industrial catalysts must be well-known, and detailed characterization of the catalyst is, therefore, required. In this study, we have produced bimetallic NiMo nanoparticles using a spark discharge generation method that produces nanoparticles of tunable, well-defined sizes and shapes. The technique has the advantage of producing clean, crystalline nanoparticles, compared to many other methods based on wet chemistry. 

An in-depth understanding of the active phase of NiMo/δ-Al_2_O_3_ will contribute to a better understanding of the catalytic reaction mechanisms involved in the hydrotreatment process. NiMo catalysts currently used for the hydrotreatment of fossil-based feedstocks could then be tailored to provide highly efficient catalysts for renewable feedstocks such as lignin.

## 2. Materials and Methods 

Industrial bimetallic NiMo catalysts were investigated and their properties compared to those of a model system of NiMo nanoparticles, produced for use in future detailed studies and optimization of the catalytic hydrotreating process of lignin. The method of production of the model and industrial catalysts, and the methods used for the characterization of both the system, are described below.

Nanoparticles with a diameter of 25 nm have a relatively low surface-to-volume ratio, resulting in particles with core atoms having properties that are different from those of the atoms at the surface. Ni and Mo are easily oxidized, and a native oxide is expected to form on the upper atomic layers of the nanoparticles upon exposure to air [15,16,17]. The NiMo in the industrial samples is expected to be fully oxidized. 

### 2.1. Spark-Discharge-Generated NiMo Nanoparticles

The nanoparticle samples were prepared using a spark discharge generator (SDG), as described elsewhere [18]. Briefly, the NiMo particles were produced using a Ni rod and a Mo rod as electrodes for the discharge sublimation of both materials, with the aim of producing completely alloyed nanoparticles. A tube furnace (operated at 1000 °C) was used to sinter the agglomerated particles formed into compact particles. H_2_ was added to the carrier gas of N_2_ (5% H_2_/N_2_) flowing at a rate of 1.68 L/min, to minimize the oxidation of the core of the particles [19,20]. A differential mobility analyzer (DMA) was used to size select monodisperse nanoparticles, and the average diameter of the selected particles was 25 nm (SD 5.5). After generation and selection, the NiMo aerosol nanoparticles were electrostatically deposited on a SiOx wafer to allow for analysis. 

### 2.2. Production of Industrial Catalysts

Three catalyst materials were prepared by incipient wetness impregnation of a δ-alumina (Al_2_O_3_) support with: (i) Ni and Mo, denoted NiMo; (ii) Ni only; and (iii) Mo only. The latter two were produced for comparison and to interpret the H_2_-TPR results obtained from the NiMo catalyst. The alumina support was crushed and screened to 10–20 mesh (0.84–2.00 mm). The first catalyst was impregnated with an aqueous solution prepared from ammonium molybdate tetrahydrate H_24_Mo_7_N_6_O_24_·4H_2_O, corresponding to 8 wt % elemental molybdenum when deposited on the support. The catalyst was aged at ambient temperature for 4 hours, dried at 120 °C for 15 hours, and calcined at 500 °C for 4 hours. This catalyst was then impregnated with nickel in an acidic aqueous solution prepared from citric acid monohydrate and nickel nitrate hexahydrate N_2_NiO_6_·6H_2_O, corresponding to 3.5 wt % elemental nickel when deposited on the support. Citric acid monohydrate was added at a molar ratio of 0.7 with respect to nickel. Again, it was aged at ambient temperature for 4 hours, dried at 120 °C for 15 hours, and calcined at 500 °C for 4 hours. This should led to a molar metal composition of 58% Mo and 42% Ni. 

The second catalyst was prepared by impregnation of alumina with an aqueous solution prepared from ammonium molybdate tetrahydrate H_24_Mo_7_N_6_O_24_·4H_2_O, corresponding to 14 wt % elemental molybdenum when deposited on the support. This is the same total metal content, on molar basis, as the first catalyst. It was then aged, dried, and calcined as described above. The third catalyst was prepared by impregnating the alumina support with nickel in an acidic aqueous solution prepared from citric acid monohydrate and nickel nitrate hexahydrate N_2_NiO_6_·6H_2_O, corresponding to 8.4 wt % elemental nickel when deposited on the support, which corresponds to the same total metal content, on a molar basis, as the first two catalysts. Citric acid monohydrate was added at a molar ratio of 0.7 with respect to nickel. It was then aged, dried and calcined as described above. These catalysts are henceforth referred to as NiMo/δ-Al_2_O_3_, Mo/δ-Al_2_O_3_, and Ni/δ-Al_2_O_3_.

## 3. Catalyst Characterization

The model NiMo nanoparticles and industrial samples were characterized in detail mainly using X-ray photoelectron spectroscopy (XPS), scanning electron microscopy (SEM), and transmission electron microscopy (TEM). In addition to the above-mentioned techniques, nitrogen physisorption (evaluated using the Brunauer–Emmett–Teller (BET) method), powder X-ray diffraction (PXRD), and H_2_-temperature-programmed reduction (H_2_-TPR) were used to obtain detailed properties of the industrial samples. 

### 3.1. X-Ray Photoelectron Spectroscopy

X-ray photoelectron spectroscopy measurements of the industrial catalysts were performed at the APXPS endstation of the SPECIES beamline at MAX IV, using a MgKα X-ray source (SPECS XR-50, SPECS GmbH, Berlin, Germany) with a photon energy of 1253 eV [21,22]. The industrial NiMo/δ-Al_2_O_3_ catalysts were ground to a powder which was drop-casted onto an Au-coated Si wafer. The charging that occurs due to insulating Al_2_O_3_ powder was minimized by performing the measurements under 1 mbar of air, and the spectra were calibrated to the Au 4f_7/2_ photoelectron peak, which has a binding energy of 84 eV. 

The model nanoparticles were investigated using an Al X-ray source with a photon energy of 1486 eV (Kratos Analytical, Manchester, UK). No charging effects were detected, and the spectra were calibrated using a Si 2p_3/2_ core level at 99.5 eV. 

When fitting the spectra, a linear background was subtracted before the components were fitted to the spectra with a Donijac Suniac lineshape convoluted with a Gaussian. Mo 3d species were fitted using a fixed spin–orbit splitting of 3.1 eV and an area ratio between the 3d_5/2_ and 3d_3/2_ components of 3:2. 

### 3.2. Transmission Electron Spectroscopy 

The nanoparticles and industrial samples were investigated at Center for High Resolution Electron Microscopy, Lund University, using a JEOL 3000F microscope (JEOL, Akishima, Tokyo, Japan), which is equipped with a field-emission gun and an energy-dispersive X-ray spectrometer (XEDS), to obtain high-resolution images and to perform compositional analysis of the samples. All samples were deposited onto lacey carbon-coated Cu TEM grids prior to analysis. 

### 3.3. Scanning Electron Spectroscopy 

The SEM study was performed at the Nano Lab at Lund University, using an FEI Nanolab 600 microscope (FEI Company, Hillsboro, OR, USA). The nanoparticles were deposited on a SiOx substrate. 

### 3.4. Powder X-Ray Diffraction 

The industrial NiMo/δ-Al_2_O_3_ catalyst was characterized using PXRD with a Stoe Stadi MP diffractometer (STOE & Cie GmbH, Darmstadt, Germany) in reflection mode. The setup has a CuKα radiation source with a Ge monochromator and a Mythen detector (DECTRIS Ltd, Baden-Daettwil, Switzerland). The diffraction pattern was recorded using a 2θ scan ranging from 15–100° with a 0.045° step size and a 55 s sampling time at each step. The amount of material in the model nanoparticles was too low to be characterized by standard PXRD measurements.

### 3.5. H_2_-Temperature Programmed Reduction and Nitrogen Physisorption

H_2_-TPR and nitrogen physisorption were performed using a Micromeritics 3-flex system (Norcross, GA, USA) operated in dynamic chemisorption and physisorption modes. Dynamic chemisorption was performed by mounting ~100 mg of the catalyst in a sample holder and passing 10 mL/min of gas (4% H_2_ in Ar) through the sample. The temperature of the sample was increased at a rate of 10 °C /min. Detection was performed using a thermal conductivity detector; water formed was removed upstream of the detector. The physisorption experiments were performed using nitrogen as the probe gas at liquid nitrogen temperatures, and the evaluation the surface areas and porous structures was performed using the desorption isotherm by the use of the methods and equations suggested by Brunauer, Emmet, and Teller (BET) [23] and by Barrett, Joyner, and Halenda (BJH) [24]. The samples were degassed at 250 °C under high vacuum for 4 h prior to the measurements.

## 4. Results

### 4.1. Characterization of NiMo Nanoparticles

The spark discharge technique allows for size selection of the nanoparticles, and by analyzing the SEM images, we were able to confirm that the particles were evenly dispersed on the SiOx substrate. The singly and doubly charged particles had size distributions centered at 25 nm and 35 nm, respectively. (Figure 1). The total standard deviation in size distribution was 5.5, but for the singly charged particles a standard deviation of 2.7 was achieved. A combination of XPS, TEM, and XEDS was performed to analyze the elementary composition of the particles, in order to confirm that alloyed NiMo particles had been formed. It was verified from the overview XPS and TEM-XEDS spectra (data not shown) that Ni and Mo were the main elements in the particles, and only a low level of carbon contamination was observed, which has been shown in previous studies to be easily removable by oxygen treatment [17]. 

A more detailed analysis of the composition of the particles was performed by STEM-XEDS mapping of the Ni and Mo in individual nanoparticles. A typical scanning transmission microscope high-angle annular dark-field (STEM HAADF) image of three nanoparticles are shown in Figure 2a. The resulting STEM-XEDS images show the integrated Mo (Figure 2b) and Ni (Figure 2c) signals, indicating that both elements are homogeneously dispersed in each nanoparticle. This is also observed in all other particles that were probed. The high spatial resolution provided by TEM allows determination of the atomic percentage of subsections of each particle using the STEM-XEDS spectra. Both Ni and Mo were observed in all 10 locations probed in 8 different particles, confirming the elemental homogeneity and the formation of a Ni-Mo alloy. More specifically, we identify that the particles contain higher levels of Ni than Mo. The average particle compositions of Ni and Mo are given in Table 1, together with the calculated Ni:Mo ratios. 

TEM images with atomic resolution were recorded, allowing the crystalline structure of the particles to be observed (Figure 2d). A Fourier transform of the TEM image revealed a lattice distance of 2.5 Å, and based on the Ni:Mo ratio found using XEDS, the investigated facet is assigned to the (2 0 0) plane in a Ni_3_Mo structure.

The Ni and Mo content was also investigated using XPS, but due to the short mean free path of the photoelectrons, the information yield is dominated by the surface of the particles. The relative elemental composition of the surface was determined by analyzing the area of the components in the fitted spectra of the Ni 2p_3/2_ and the Mo 3d regions. The program CASA XPS was used in the evaluation of the spectra. The detection probability for the photoelectrons from Ni and Mo (detector transmission efficiency, i.e., the relative sensitive factor) and the mean free path of the electrons were taken into consideration. The Ni:Mo ratio on the surface of the particles was determined to be 2.6. The ratio obtained from XPS is higher than that determined using XEDS, indicating that there is an enrichment of Ni in the surface as compared to the core, of the particles (Table 1).

The chemical composition of the surface atoms are essential for the activity of the catalyst and can be obtained by careful spectral analysis of the Ni 2p, O 1s, and Mo 3d regions. Several species are resolved in the spectrum from the Mo 3d core level (Figure 3a), but fitting components to them is complex due to the overlap of chemical shift and spin-orbit split effects [25]. Three oxidation states of Mo were revealed by maintaining the ratio of the peak areas to 3:2 of the spin-orbit peaks that are separated by 3.1 eV and these were assigned to Mo^0^, Mo^4+^, and Mo^6+^. The Mo^0^ components (228.44 eV [Mo 3d_5/2_] and 231.57 eV [Mo 3d_3/2_]) observed in the spectrum clearly indicate the presence of metallic Mo in the nanoparticles [26,27]. Since Mo is easily oxidized in air, we propose that the metallic species originates from the core of the particles, and the other two species from a native oxide that forms as a self-limited shell on the particles. The information depth of the XPS signal (based on the assumption that the detected electrons originate from 3 x the inelastic mean free path) is in the range of 3–5 nm; therefore, we conclude that the native oxide is less than 5 nm thick. The first oxide component fitted to the spectrum has binding energies of 229.51 eV [Mo 3d_5/2_] and 232.62 eV [Mo 3d_3/2_] (yellow in the Mo 3d spectrum in Figure 3a) and is assigned to Mo^4+^, corresponding to MoO_2_ oxide formation [25,28,29]. The second oxide component is shifted to an even higher binding energy, which suggests that Mo is also in a higher oxidation state, and is fitted with the components at a binding energies of 232.71 eV [Mo 3d_5/2_] and 235.81 eV [Mo 3d_3/2_] (green in the Mo 3d spectrum in Figure 3a). This component is interpreted as electrons originating from a Mo^6+^ species [26], which suggests the presence of either a NiMoO_4_ alloy [30,31,32] or MoO_3_ [25,28]. The component corresponding to the Mo^4+^ species, however, does not show a perfect fit to the measured spectrum, but fulfills the criteria of a peak area ratio of 3:2 between the Mo 3d_5/2_ and Mo 3d_3/2_ components and the spin-orbit split. We attribute this discrepancy between the fit and the measured data to an unresolved effect, which contributes to an uncertainty in the results given in Table 1.

Alloy formation is also evident in the Ni spectrum (Figure 3c). Two sharp peaks, as well as a broad feature, are observed in the Ni 2p_3/2_ region. The first sharp peak at 853.20 eV is observed at ~0.5 eV higher binding energy than tabulated for metallic Ni (852.7 eV [33]), but we assign this component to metallic Ni (Ni^0^) due to the presence of metallic Mo. The second component is observed at 856.0 eV, and is in good agreement with literature values for a Ni^2+^ species in NiMoO_4_ [34,35]. The Ni^2+^ species may also originate from Ni_2_O_3_ due to the high Ni content of the particles. The broad component in the Ni 2p_3/2_ region is believed to be a Ni^2+^ satellite feature, in agreement with previous observations [34]. 

It is clear from the O1s spectrum (Figure 3b) that at least two O species are present on the surface. The main peak, at a binding energy of 533.18 eV, is assigned to SiOx from the substrate [36], while the component at 531.12 eV is assigned to oxygen associated with the oxide shell of the nanoparticle as a mixture of MoO_3_ [26] and NiO [37]. 

In summary, we conclude that at the probing depth of XPS, both metallic and oxide phases are present in the nanoparticles. A composition of 72 atomic % Ni and 28 atomic % Mo was determined from XPS, which can be compared to the composition of 66 atomic % Ni and 34 atomic % Mo (SD 8.4) observed using XEDS, which is a more bulk-sensitive technique. According to the XPS data approximately 42% of the Mo is metallic, and the remaining 58% is oxidized, where the higher oxidation state (Mo^6+^) is slightly more prevalent than Mo^4+^. The chemical shifts of pure Mo oxides are similar to those of the alloyed NiMo oxides, but it is difficult to determine from the XPS data alone whether a NiMo alloy is formed at the surface. However, XEDS mapping of Ni and Mo indicates that the metals are well mixed, and that the amount of Mo is significantly less than the amount of Ni. Hence, the formation of MoO_3_ is less likely, and we conclude that the chemical shifts observed in the Mo 3d spectrum are due to NiMo alloy oxide formation. This is also in good agreement with the Ni spectrum, where the oxidized Ni corresponds to approximately 60% of the probed Ni.

### 4.2. Characterization of the Industrial NiMo/δ-Al_2_O_3_ Catalyst

The industrially produced NiMo/δ-Al_2_O_3_ catalyst used in the hydrotreatment process of lignin-based feedstock was characterized in detail to reveal the key properties that should be reflected in the simplified model system. The δ-phase of Al_2_O_3_ is used as a porous support to generate a large surface area over which the Ni and Mo is dispersed. The specific surface area (SBET) of the NiMo/δ-Al_2_O_3_ was estimated to be 109 m^2^/g. Calculations using the BJH method gave a pore volume of 0.53 cm^3^/g, with an average pore size of 16.5 nm, indicating that the material has a mesoporous character. 

A procedure similar to that described for the nanoparticles was applied to analyze the chemical composition of this industrial catalyst. It is important that the active materials in an industrial catalyst are homogeneously and well-dispersed over the support. XEDS was used to map the Ni, Mo, Al, and O contents (Figure 4). 

Elementary analysis was performed at 26 different positions (10 × 10 nm^2^) on the industrial sample using STEM-XEDS, and the atomic percentage of Al, O, Ni, and Mo was determined. The Ni:Mo ratio is highly relevant for the activation of the catalyst, and the average atomic percentage of the active metals was calculated to be 29% Ni and 71% Mo (SD 2.9), resulting in an average Ni:Mo ratio of 0.4 (Table 2). The homogeneous dispersion of Ni and Mo indicates that the Ni and Mo are well mixed, and that an alloy is probably formed. 

PXRD was also used to further investigate whether a NiMo alloy was formed. The intense, narrow diffraction peaks identified in the 2θ scan confirm the presence of the δ phase of the Al_2_O_3_ (Figure 5). A diffraction peak can also be observed around 27° and is assigned to the NiMoO_4_ structure [34,38]. It should be noted that a crystalline structure is necessary to give rise to a diffraction pattern, but the broad, weak diffraction peak indicates that the amount of crystalline NiMoO_4_ particles is low, or that the particles are small. This may be explained by the good dispersion of Ni and Mo, where the strong interaction with the support prevents Ni or Mo from forming a crystalline structure. Based on the above results, and the TEM results where no crystalline structures could be observed, we conclude that an amorphous structure of Ni and Mo dominates the industrial catalyst.

Despite the fact that a well-ordered alloy structure is not formed, the interaction between Ni and Mo is important for the activity of the catalyst, as can be shown using a combination of XPS and H_2_-TPR to probe the chemical information of the catalyst (Figure 6). The XPS study of the insulating industrial samples was conducted under 1 mbar of air, showing that the charging effect that occurs can be effectively reduced by the presence of a gas [39]. Two peaks can be resolved in the Mo 3d XPS spectrum at binding energies of 233.1 eV and 236.2 eV (Figure 6b). Similar peaks were found in the Mo 3d region in the investigation of the NiMo nanoparticles, and thus the peaks are assigned to the Mo^6+^ oxidation state with a spin-orbit splitting of 3.1 eV. As mentioned above, peak assignment is not trivial, and the Mo^6+^ component may be correlated to the NiMoO_4_ alloy, pure MoO_3_, or a molybdate in Al_2_(MoO_4_)_3_ [40,41,42]. A component is observed in the Ni 2p_3/2_ spectrum at a binding energy of 856.0 eV, similar to the nanoparticle, and is accordingly assigned to a Ni^2+^ species. It should be noted that no metallic Ni or Mo is observed in the XPS spectra. Based on the XPS results in combination with the PXRD data, we can assign the Ni^2+^ and Mo^6+^ species to the NiMoO_4_ alloy [43]. The Ni:Mo ratio was calculated using the same procedure as for the nanoparticles, resulting in a value of 0.3, which is in good agreement with the ratio calculated from XEDS (0.4). The ratio determined by XPS is expected to be similar to that determined by XEDS as the well-dispersed Ni and Mo do not give rise to any significant difference in the ratio between the surface and bulk, as was observed for the nanoparticles. 

The interaction between Ni and Mo in the NiMo/δ-Al_2_O_3_ was further confirmed using H_2_-TPR (Figure 6c). Industrial catalysts containing only Ni (Ni/δ-Al_2_O_3_) or Mo (Mo/δ-Al_2_O_3_) were investigated for comparison, and to help interpret the results obtained from the catalyst containing both Ni and Mo. The reduction of Ni/δ-Al_2_O_3_ takes place over a broad temperature range, but significant reduction is observed at 600 °C, which is approximately 200 °C higher than that reported for NiO [44]. This high reduction temperature is an indication of a strong interaction between NiOx and δ-Al_2_O_3_ [45]. The Mo/δ-Al_2_O_3_ catalyst shows two distinct reduction peaks, at 540 °C and 960 °C. Reduction at 540 °C is interpreted as the partial reduction of amorphous molybdates or MoO_3_, where Mo is in an octahedral oxygen structure (Mo^6+^ → Mo^4+^) [44,46]. Reduction at higher temperatures, between 900 °C and 1100 °C, is believed to be further reduction of Mo (Mo^4+^ → Mo^0^) [34]. The high reduction temperature suggests that this Mo species interacts strongly with the Al support. 

Interestingly, the main reduction of the bimetallic NiMo/δ-Al_2_O_3_ catalyst occurs at a significantly lower temperature than in the Ni/δ-Al_2_O_3_ or Mo/δ-Al_2_O_3_. Reduction of Mo in the NiMo/δ-Al_2_O_3_ starts at a temperature approximately 100 °C lower than in Mo/δ-Al_2_O_3_ (Mo^6+^ → Mo^4+^). Also, the second reduction stage of Mo (Mo^4+^ → Mo^0^) is shifted towards a lower temperature. A minor reduction is also observed at 750 °C, which can be assigned to the reduction of Ni/δ-Al_2_O_3_ [47]. We therefore conclude from the results of the TPR experiments that the addition of Ni to the catalyst promotes the reducibility of Mo, and is a clear marker of the interaction between Ni and Mo in the sample. The decreased reduction temperature of Mo in the NiMo catalyst as compared to the Mo/δ-Al_2_O_3_ or Ni/δ-Al_2_O_3_ catalysts has also been observed by L. Qu et al. for the same active metals supported on γ-Al_2_O_3_ [44].

## 5. Discussion

Nanoparticles are often produced and used as model catalysts to gain a fundamental understanding of chemical processes, but there is often no comparison between the model system and industrially employed catalysts. These model systems may, therefore, lack relevance for current industrial applications. In this study, we have demonstrated how the properties of both an industrial catalyst and a potential the model system can be determined using a combination of bulk- and surface-sensitive techniques (Figure 7). 

We have successfully produced 25 nm NiMo bimetallic nanoparticles using the spark discharge method, which allows for size selectivity of the particles. Measurements of the metal composition of the particles using XEDS and XPS show that the particles contain more Ni than Mo. This may be due to segregation of Ni to the surface when the particles are exposed to air, but further studies are needed to confirm potential segregation effects. The XPS results also indicate oxide formation on the outer atomic layers.

In the production of the industrial catalyst, a molar content of 58% Mo and 42% Ni are expected based on the recipe described in the Materials and Methods. Our results, however, indicate 70% Mo and 30% Ni, which may be due to losses of Ni in the impregnation and calcination processes. Using XEDS we concluded that the metals are well-dispersed on δ-Al_2_O_3_ support, which is ideal from an industrial point of view. The uniform dispersion of Ni and Mo metals is also supported by the weak PXRD signal originating from crystalline NiMoO_4_ alloy, and by the TEM images where no clear crystalline structure of the metals could be observed. The H_2_-TPR results provide clear evidence that Ni promotes the reduction of Mo oxide, and confirm that there is an essential interaction between the Ni and Mo atoms, which changes the chemical properties of the two metals. This is also corroborated by the XPS results. The strong interaction between Ni and Mo is, therefore, an essential property that should be considered in the design of a model catalyst. 

Further studies are required on these model nanoparticles to tune their properties so that they better reflect industrial catalysts. A lower Ni:Mo ratio is required to improve the model system, which could be achieved by changing the parameters such as the furnace temperature and the gas flow in the production of the nanoparticles. Kala et al. have also shown that the alloy composition can be controlled by changing the charging current if pure Ni and Mo are used as electrodes in the spark discharged generator [48]. An alternative would be by using alloyed electrodes in the set up to tailor the metal composition. The model system can also be further tuned by decreasing the size of the nanoparticles and narrow the size distribution, which can be accomplished by adding another DMA in production. Moreover, a comparison of the activity in the hydrotreatment process of lignin between the industrial catalyst and our model system should be performed to verify that our model catalyst can be used to increase the fundamental understanding of the process.

## 6. Conclusions

To conclude, the results of this study show that we successfully produced bimetallic nanoparticles. Our findings demonstrate that the spark discharge method generates alloyed NiMo nanoparticles with a metallic core surrounded by an oxide shell. The oxidation state of the metals in the surface of the nanoparticles determined by XPS is in good agreement with the oxidation states found for the metals in the industrial catalyst. The active metals in the industrial catalyst were well-dispersed over the support and our TPR and XPS results indicate that the Ni and Mo atoms are interacting which promotes the reduction temperature of the NiMo catalyst. Although further studies are needed to tune the properties of the nanoparticles, our results indicate that the particles are well suited for being used as a simplified model of complex industrial NiMo/δ-Al_2_O_3_ catalysts. Such a model system will allow us to gain a better understanding of the catalytic hydrotreatment process, which is particularly important in the valorization of biomass. 

## Figures and Tables

**Figure 1 materials-12-03727-f001:**
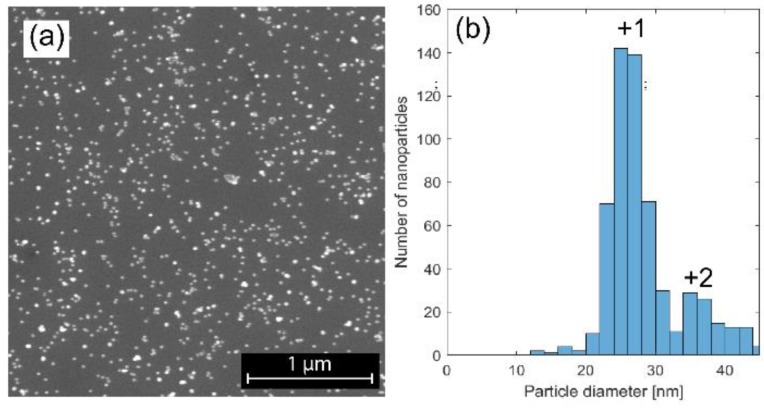
SEM image of the nanoparticles with the graph showing the size distribution. (**a**) The SEM image, showing nanoparticles evenly dispersed on the SiO_x_ substrate; (**b**) The size distribution of the particles selected by the DMA. Singly charged particles that pass through the DMA have a diameter of 25 nm, while a small fraction of doubly charged particles has a diameter of 35 nm.

**Figure 2 materials-12-03727-f002:**
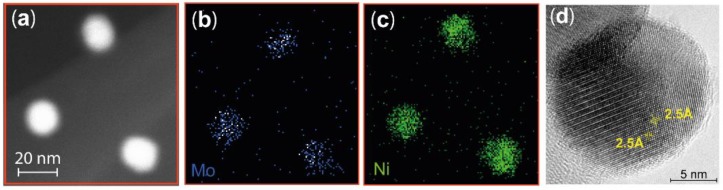
(**a**) Typical STEM HAADF image of three nanoparticles. An example of STEM XEDS mapping of Mo (**b**) and Ni (**c**) indicating that the Ni and Mo are homogeneously mixed in all three nanoparticles. The scale is the same in (**a**–**c**). (**d**) The high-resolution TEM image of one nanoparticle shows that the nanoparticles have a crystalline structure with a lattice distance of 2.5 Å.

**Figure 3 materials-12-03727-f003:**
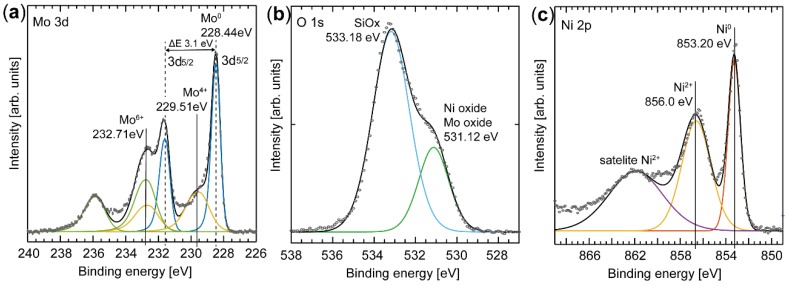
XPS spectra of Ni 2p_3/2_, O1s, and Mo 3d. The fits indicate that Ni and Mo are mixed, forming an alloy. (**a**) The Mo 3d spectrum is complex and the fitted components are color-coded to account for the spin-orbit splitting, and only the binding energy of the 3d_5/2_ component is indicated. The peaks at 228.44 eV and 231.57 eV (blue) originate from metallic Mo. The peaks at 229.51 eV and 232.62 eV (yellow) are interpreted as originating from Mo^4+^. The most shifted component in binding energy at 232.71 eV and 235.81 eV (green) is assigned to Mo^6+^. (**b**) The O1s spectrum has two clear components assigned to the SiOx in the substrate and the oxide in the particles. (**c**) The Ni 2p_3/2_ spectrum is fitted with three components, where the peak at 853.2 eV originates from metallic Ni, while the component at 856 eV is assigned to the oxide in the NiMoO_4_ alloy.

**Figure 4 materials-12-03727-f004:**
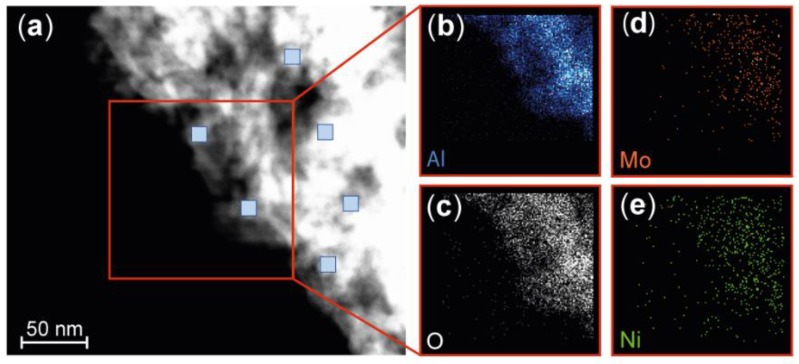
STEM HAADF and XEDS mapping of the industrial NiMo/δ-Al_2_O_3_ catalyst. (**a**) STEM HAADF image at one edge of the catalyst. The blue squares indicate the region in which elemental analysis was performed. The scale is the same for all images. Strong Al (**b**) and O signals (**c**) were observed in the XEDS mapping image, originating largely from the alumina support. The much weaker signals from Mo (**d**) and Ni (**e**) are due to the low concentrations of these metals in the catalyst, but XEDS mapping indicates that the catalytically active Ni and Mo are well-dispersed over the support. The loading of Ni and Mo is estimated to be submonolayer on the δ-Al_2_O_3_ surface, based on the measured wt % of the Ni and Mo and the surface area of the support.

**Figure 5 materials-12-03727-f005:**
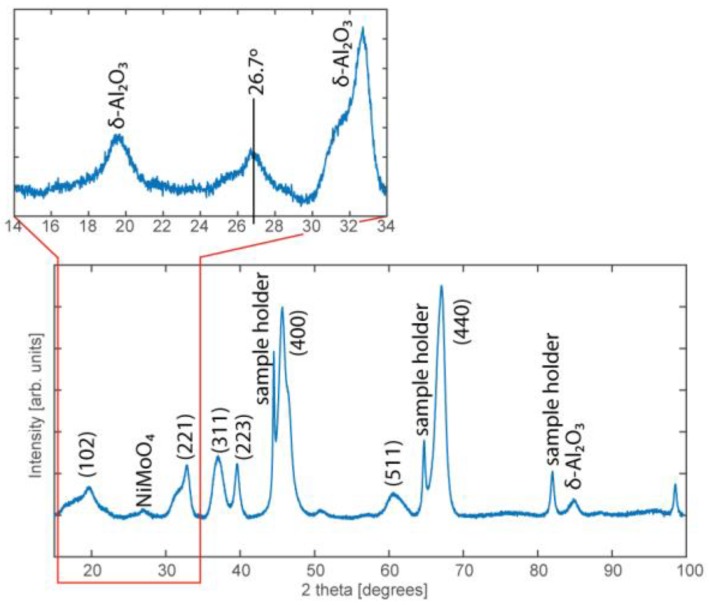
PXRD diffraction pattern obtained from the industrial NiMo/δ-Al_2_O_3_ catalyst where the majority of the peaks originate from planes of the δ-Al_2_O_3_ support (indicated by hkl indices). A broad, weak diffraction peak is, however, also visible at 26.7° which corresponds to the NiMoO_4_ structure.

**Figure 6 materials-12-03727-f006:**
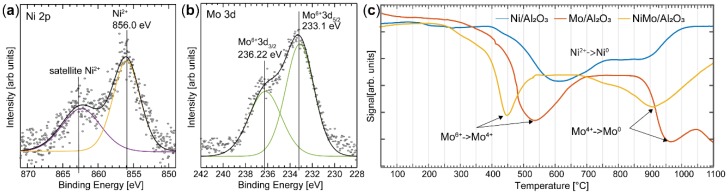
Chemical analysis of the NiMo/δ-Al_2_O_3_ sample was performed using XPS and H_2_-TPR. (**a**) XPS spectra of the Ni 2p_3/2_ and (**b**) Mo 3d regions. Fitting the spectra indicates a strong interaction between Ni and Mo. (**c**) H_2_-TPR spectra from the Mo/δ-Al_2_O_3_, Ni/δ-Al_2_O_3_, and NiMo/δ-Al_2_O_3_ catalysts. The results show that Ni promotes the reduction of the Mo oxide.

**Figure 7 materials-12-03727-f007:**
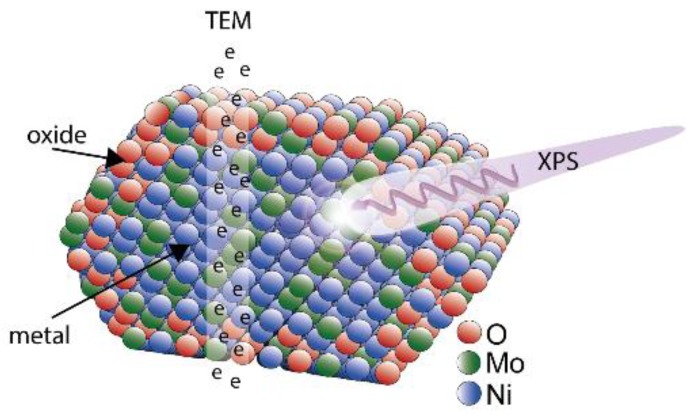
A schematic of the NiMo nanoparticle produced, where the high energy of the electrons in TEM-XEDS probe a cross-section of the particle, while XPS provides information of the surface. The XPS results indicate oxide formation on the outer atomic layers with a higher Ni content than in the core of the particles.

**Table 1 materials-12-03727-t001:** Average atomic percentage of Ni and Mo calculated from the STEM-XEDS spectra and the XPS spectra from the particles

Technique	Ni (Atomic %)	Mo (Atomic %)	Ni:Mo
STEM-XEDS	66	34	1.9
XPS	72	28	2.6

**Table 2 materials-12-03727-t002:** Average atomic percentage of Ni and Mo in the industrial NiMo/δ-Al_2_O_3_ catalyst measured at 26 different positions on the support using STEM-XEDS, and the Ni and Mo composition determined by XPS.

Technique	Ni (Atomic %)	Mo (Atomic %)	Ni:Mo
STEM-XEDS	29	71	0.4
XPS	23	77	0.3

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
