# Peer review of "Bimetallic Nanoparticles as a Model System for an Industrial NiMo Catalyst"

_materials, 2019, doi:10.3390/ma12223727_

Round 1

Reviewer 1 Report

The paper is as a whole fine, well understandable, interesting and presenting significant novelties and an accurate and I think fully objective report of the results obtained. I only suggest to present the Conclusions (after "To conclude") in a separate section, possibly extending them a bit, illustrating very shortly the whole of the results obtained.

Reviewer 2 Report

The article entitled »Bimetallic nanoparticles as a model system for an industrial NiMo catalyst« by S. Blomberg et al. represents an experimental and characterization study, where NiMo nanoparticles were prepared by a spark discharge generation method and compared with industrial NiMo catalyst prepared by wetness method. The catalysts are well characterized, the results are properly interpreted, but at least the determination of their activity are lacking to evaluate the performance of the obtained catalysts.  I would consider this work suitable for publication after a major revision, after the points (listed below) are adequately addressed.

Comment #1: It is not clear from the current results whether the catalyst obtained by the spark-discharged method is active or not. The authors should determine the activity of the catalyst and compare it with the industrial catalysts or the reaction system studied in the literature should be tested at exactly the same conditions.

Comment #2: The role of Ni, Mo, NiMoSx and MoS2 sites were studied experimentally and computationally in the literature. These works should should be cited and commented (DOI: 10.1016/j.jcat.2011.01.025, 10.1016/j.apcatb.2017.06.046, …).

Comment #3: The authors observed the presence of carbon impurities in the catalyst (line 168). How the presence of impurities could affects its activity? A comparison with the relevant literature is desirable.

Comment #4: The authors claim that the presence of Ni on the catalyst promotes the reducibility of Mo (line 306). It should be explained how it is promoted and the claim must be supported by appropriate literature.

Comment #5: Lines 316 to 320 are the same as the title of Figure 7. The authors should correct accordingly.

Comment #6: In the discussion section (line 334 to line 344) the authors propose further studies to be included. However, at least some of the proposed improvements should be added to the current study and above all, to test activities that will confirm or disprove the supremacy of the proposed catalyst synthesis method.

Round 2

Reviewer 2 Report

Authors carefully reviewed the manuscript according to the questions raised and applied all the required corrections. The manuscript is now suitable for publication in Materials.